# (GIGA)byte

TECHNICAL RELEASE

# Nanopore-based enrichment of antimicrobial resistance genes – a case-based study

Adrian Viehweger[1,*], Mike Marquet[2], Martin Hölzer[3], Nadine Dietze[1], Mathias W. Pletz[2] and Christian Brandt[2]

1 Institute of Medical Microbiology and Virology, University Hospital Leipzig, Leipzig, Germany
2 Institute for Infectious Diseases and Infection Control, Jena University Hospital, Jena, Germany
3 MF1 Bioinformatics, Robert Koch Institute, Berlin, Germany

## ABSTRACT

Rapid screening of hospital admissions to detect asymptomatic carriers of resistant bacteria can prevent pathogen outbreaks. However, the resulting isolates rarely have their genome sequenced due to cost constraints and long turn-around times to get and process the data, limiting their usefulness to the practitioner. Here we used real-time, on-device target enrichment ("adaptive") sequencing as a highly multiplexed assay covering 1,147 antimicrobial resistance genes. We compared its utility against standard and metagenomic sequencing, focusing on an isolate of *Raoultella ornithinolytica* harbouring three carbapenemases (*NDM, KPC, VIM*). Based on this experimental data, we then modelled the influence of several variables on the enrichment results and predicted the large effect of nucleotide identity (higher is better) and read length (shorter is better). Lastly, we showed how all relevant resistance genes are detected using adaptive sequencing on a miniature ("Flongle") flow cell, motivating its use in a clinical setting to monitor similar cases and their surroundings.

**Subjects** Microbiology, Medical Microbiology, Bioinformatics

**Submitted:** 29 November 2022

\* Corresponding author. E-mail: adrian.viehweger@medizin. uni-leipzig.de

Preprint submitted at https: //doi.org/10.1101/2021.08.29.458107

## BACKGROUND

Screening patients for multiresistant bacteria on hospital admission can detect asymptomatic colonization early [1] and reduce subsequent complications [2]. However, corresponding isolates rarely have their genome sequenced, which would enable genomic surveillance, and, as a result, source control and reduced spread [3]. Such resistant strains can colonize patients for years, increasing the value of this information [4]. Long-term carriage is surprising in the absence of a selective stimulus such as treatment with antimicrobials. Recently, the underlying microbial consortia in which these strains are embedded have been implicated in resistance maintenance through ongoing horizontal gene transfer of mobile elements [5, 6]. This finding suggests that, in special cases, genomic surveillance should be expanded to include metagenomic data [7].

Here we report on a patient with multiple carbapenem-resistant strains detected in a rectal swab. One of the isolates simultaneously carried three carbapenemases, an unusually high number. To support a timely response, we integrated the results from multiple modalities of real-time nanopore sequencing. First, we reconstructed the genomes of individual isolates and then complemented them with metagenomic data from the swab. In a proof-of-concept, we then applied real-time on-device target enrichment of 1,147 resistance genes on a miniature flow cell [8] to create an ultra-high multiplex assay.

## METHODS

### Culture and DNA extraction

All samples were streaked on carbapenemase chromogenic agar plates (CHROMagar, Paris, France). Carbapenemase carriage was confirmed using PCR and phenotypically using microdilution minimal inhibitory concentration testing. DNA was extracted from culture isolates and rectal swabs using the ZymoBIOMICS DNA Miniprep extraction kit according to the manufacturer's instructions. The cell disruption was conducted three times for five minutes with the Speedmill Plus (Analytik Jena, Germany).

### Library preparation

DNA quantification steps were performed using the dsDNA HS assay for Qubit (Invitrogen, US). DNA was size-selected by cleaning up with 0.45× volume of Ampure XP buffer (Beckman Coulter, Brea, CA, USA) and eluted in 60 µl EB buffer (Qiagen, Hilden, Germany). The libraries were prepared from 1.5 µg input DNA. For multiple samples, we used the SQK-LSK109 kit (Oxford Nanopore Technologies, Oxford, UK) and the Native Barcoding Expansion-Kit (EXP-NBD104), according to the manufacturer's protocol. For the Flongle run, we used the SQK-RBK004 kit from the same manufacturer.

### Nanopore sequencing and on-device target enrichment

All DNA was sequenced on the GridION using FLO-MIN106D (MinION) and FLO-FGL001 (Flongle) flow cells (`MinKNOW` software v4.1.2), all from Oxford Nanopore Technologies. Data on sequencing statistics for all runs are provided in the project code repository [9]. Three sequencing runs were performed: for the first run (MinION flow cell), we multiplexed three culture isolates (A2, B1, B2) and a metagenomic sample (3.9 M reads and 23.3 Gb in 48 h, about 10 Gb were metagenomic). The second run (MinION flow cell) was an experiment comparing "adaptive" and "standard" sequencing. On a single flow cell, we periodically alternated between both states by manually turning the sequencing run off and then back on in the other state in one-hour intervals for a total of 16 hours. Toggling between states did not harm sequencing (e.g., through pore blockages; 3.44 M reads and 4.47 Gb in 16 h). The sequencing yield for all barcodes from three isolates (two *Citrobacter* and one *Raoultella*) with two technical replicates each was about equal. Adaptive sequencing groups the sequence data into "rejected", i.e., reads that do not contain a target, and "unrejected". The latter comprises reads with a target found and reads with a pending decision. We used the unrejected reads pooled across isolates and replicates for all further analyses unless stated otherwise. Pooled and per-isolate results did not differ substantially (compare Figures 1 and 5B). In the third sequencing run (Flongle flow cell), only isolate A2 was included, and adaptive sampling was applied throughout (18,646 reads and 5.36 Mb in 4 h).

As target database, we created a dereplicated version of the *CARD* database of resistance genes (v3.1.3) [10] using `mmseqs2 easy-cluster` (v13.45111) [11] using a minimum sequence identity of 0.95 and minimum coverage of 0.8 in coverage mode 1. We thereby reduced the database from 2,979 to 1,147 representative genes. We performed this step to reduce the search space that the adaptive sequencing algorithm has to map against. The total length of all genes in the database was 1.16 Mb. The reduction halves the database size because many resistance genes, such as *CTX,* have many documented isoforms, which would lead to uninformative multi-mappings. Reads were basecalled using the `guppy` GPU basecaller (high accuracy model, v4.2.2, Oxford Nanopore Technologies). For isolate genomes, reads were

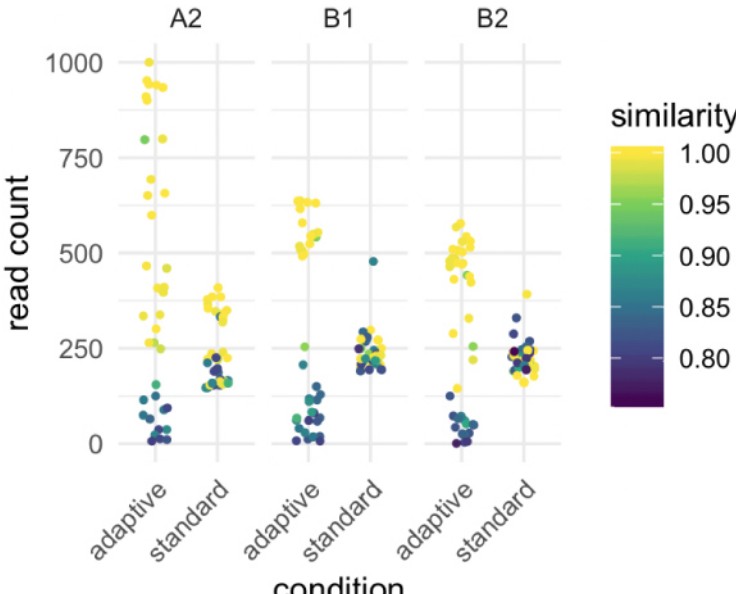

**Figure 1.** While the main manuscript analyses the pooled reads from three isolates, we here demonstrate that the effects observed in aggregate reproduce at the level of the individual isolates (compare Figure 5B).

assigned to their respective barcodes only if matching adapters were detected on both ends of the read to avoid cross-contamination.

For the experiment comparing "adaptive" and "standard" sequencing on a single flow cell, we periodically alternated between both states by manually turning the sequencing run off and then back on in the other state using one-hour intervals for a total of 16 hours. This protocol did not have a negative effect on the total sequencing yield over time (e.g., through pore blockages). The sequencing yield for all barcodes from three isolates (two *Citrobacter* and one *Raoultella*) with two technical replicates each was about equal. Adaptive sequencing groups the sequence data into "rejected", i.e., reads that do not contain a target, and "unrejected". The latter comprises reads where a target has been found and reads with a pending decision. We used the unrejected read fraction for all further analyses.

## Data analysis

Isolate data were assembled using `flye` (v2.9, RRID:SCR_017016) [12] and consensus sequences corrected using three rounds of polishing with `racon` (v1.4.3, RRID:SCR_017642) [13] followed by `medaka` (v1.4.3, unpublished) [14]. Read mapping was performed using `minimap2` (v2.22-r1101, RRID:SCR_018550) [15]. Genome quality was confirmed using `checkm` (v1.1.3) [16]. All isolate genome assemblies were >99% complete and <1% "contaminated" (duplicate single-copy marker genes), which counts as high quality by community standards [16]. Resistance gene annotation was performed using `abricate` (v1.0.1, unpublished, github.com/tseemann/abricate) against the *CARD* database (see above). Taxonomic assignments were performed using single-copy marker genes [17] as well as k-mers using sourmash (v4.2) [18].

Human DNA sequences were removed from the metagenomic stool dataset before analysis by filtering them against the recently published complete human reference



genome CHM13 [19] using `minimap2` [15] with default settings. For the long read-only metagenomic assembly we used `flye` with the `--meta` flag. We then mapped all reads to the assembly using `minimap2`. We then used `racon` to perform three rounds of long read-only polishing of the assembly using the alignment. Last, we used `medaka` to generate the final consensus assembly. Binning and annotation were then performed, as described elsewhere [20], by feeding the consensus assembly into the corresponding workflow modules using default settings. Pairwise similarity between genes was calculated using `mmseqs2` `easy-search` (v13.45111) [11]. The amount of putative horizontal gene transfer between isolate genomes and metagenome-assembled genomes (MAGs) was estimated by counting the number of shared genes for each pair. First, we performed pairwise genome alignment using the `nucmer` command from `mummer` (v4.0.0rc1) [21]. We then searched for "shared genes", defined as such if the alignment was 1 kb or longer and if the pairwise nucleotide identity between genes was >99.9%. Nucleotide identity here is defined as in `mummer4` [21].

To model the influence of several predictor variables on our outcome variable "target abundance" (total on-target read count), including plausible interactions, we fit a Bayesian regression model using `brms` (v2.13) [22]. The outcome variable was modeled as a Poisson distribution. We conditioned the effect of nucleotide similarity on sequencing state, i.e., whether adaptive sequencing was turned on or off, by introducing an interaction term. Also, we conditioned read length on sequencing state and whether a read was derived from a plasmid or the chromosome. Finally, we included a term to model the effect of contig coverage, calculated from mapping the reads back to the isolate assemblies. Note that here nucleotide identity is defined as in `mmseqs2` (v13.45111, RRID:SCR_008184) [11].

$$
\begin{aligned}
read\_count &\sim Normal(\mu_i, \sigma) \\
\mu_i &= \alpha + \gamma_i adaptive_i + \delta_i plasmid_i + \delta_i adaptive_i + \beta_1 coverage \\
\gamma_i &= \beta_2 + \beta_3 similarity \\
\delta_i &= \beta_4 + \beta_5 log(read\_length)
\end{aligned}
$$

Sampling was performed with four chains, each with 2,000 iterations, of which the first half were discarded as warmup, for a total of 4,000 post-warmup samples. Samples were drawn using the NUTS algorithm. All chains converged ($R$ = 1.00).

For the simulation experiment, we first determined that a log-normal distribution can adequately model the different length distributions (Figure 2). We then selected parameters to model combinations of reads and targets of varying length distributions, from long (mean 8,103 bases, or log 9, variance 1.5) to short (mean 665 bases, or log 6.5, variance 0.25). We could thus, for example, assess the effects of combining long reads with short targets and short reads with long targets. Next, we generated ten thousand (read, target) sample pairs for each combination of read and target distributions. For each pair, we randomly "placed" a target start on the read with a uniform distribution across read positions, in line with how targets are distributed across DNA fragments in realistic single-molecule sequencing experiments. We then asked if the first part of the read (as seen by the adaptive sampling algorithm) would have detected the target. Since all simulated reads contain a target, failure to detect one in the first number of bases counts as a false negative (Figure 3). To estimate effect sizes on the false-negative rate (FNR), we fit a multivariate regression



model using `brms` (see above) (Figure 3):

$$FNR \quad \sim \quad Normal(\mu_i, \sigma)$$
$$\mu_i \quad = \quad \alpha + \beta_1 log(read\_length) + \beta_2 log(target\_length)$$

## RESULTS

## A case of extensive antimicrobial resistance is characterized using isolate and metagenomic Nanopore-sequencing

During the resistance screening of rectal swabs, we found three bacterial species growing on carbapenem agar (*Raoultella ornithinolytica*, *Citrobacter freundii*, and *Citrobacter amalonaticus*). The patient's history revealed no apparent source, although past occupations included work in waste management and training in agriculture, both of which have increased exposure to antibiotic resistance genes [23]. Surprisingly, we detected multiple carbapenemases in *R. ornithinolytica* using PCR (*NDM, KPC, VIM*). To identify all the resistance genes in the isolates and any putative horizontal transfer between them, we performed real-time nanopore sequencing, both of the isolates individually and of the entire rectal swab, generating in total 3.9 M reads and 23.3 Gb on a standard ("MinION") flow cell.

The *R. ornithinolytica* isolate carried nine plasmids and three carbapenemases: *NDM-1*, *KPC-2*, and *VIM-1* (Figure 4A). All carbapenemases were encoded on one plasmid each, except *VIM*, which was located on the bacterial chromosome.

The two *Citrobacter* isolates only carried *VIM-1*. An alignment of the genomic region 10 Kb upstream and downstream of *VIM* across the isolates revealed a transposase-mediated resistance transfer, for which we propose the following gene flow: The genomes of *C. freundii* and *C. amalonaticus* both carry *VIM-1* on an *IncHI2* plasmid (>95% sequence identity). In *C. freundii*, this transposon then likely copied itself into an *IncN* plasmid with the help of an *ISKpn19* transposase (Figure 4B). The same transposase is found flanking the *VIM* transposon in the *R. ornithinolytica* chromosome, which makes the *IncN* plasmid of *C. freundii* its likely source. A similar transfer pattern was observed for the penicillinase *OXA-1* (data not shown).

Isolate sequencing captured 85.5% (59/69) of resistance genes detected in the underlying microbial consortium through metagenomic sequencing (total yield 10 Gb, Figure 4C). Of the remainder, few genes were clinically relevant, such as several efflux pumps. Other resistance genes were associated with Gram-positive bacteria, which we did not screen for with culture (Figure 4C). Surprisingly, metagenomics did not detect five resistance gene types, including *KPC*, two out of three *OXA* copies, and two out of four *VIM* copies. This omission likely occurs because the metagenome was dominated by *Proteus vulgaris* (44.6% of reads), leaving fewer reads (depth) for the carbapenemase-carrying strains (*C. freundii* 19.7%, *R. ornithinolytica* 1.8%, *C. amalonaticus* 0.01%). Selective culture enriched these low-abundant species.

We also observed substantial horizontal gene transfer between our isolate members of the *Enterobacteriaceae* (Figure 4D). For example, *C. freundii* and *R. ornithinolytica* share 15 loci. A region was labeled as a putative transfer if its length exceeded one kilobase with 99.9% sequence identity between any two genomes. No additional transfer was found in two uncultured MAGs, namely *Enterococcus faecium* and *Serratia ureilytica*. None of the

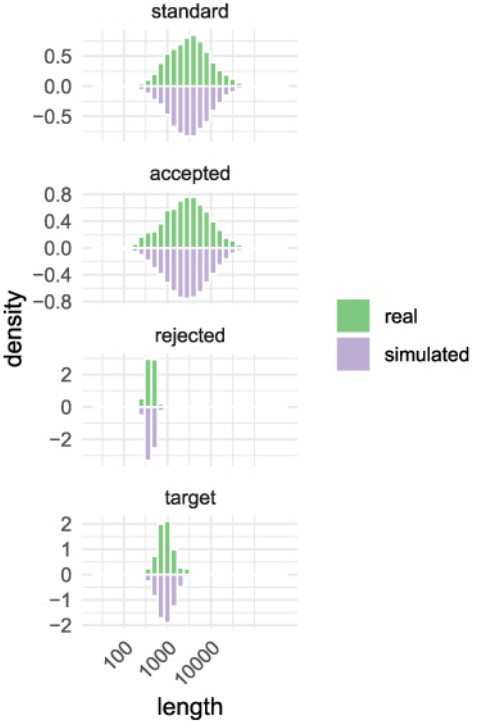

**Figure 2.** Exemplary length distribution of various sequence sets of interest discussed in the study (isolate A2). Generated reads and target genes in green; simulated sequences in violet. Note that the x-axis uses the log scale. As expected, reads generated using standard sequencing did not differ in their length distribution from unrejected reads from adaptive sampling. Only the sequences of rejected reads were truncated, usually after, on average, 415 bases (median). The antimicrobial resistance genes in our target database had a mean length of 1010 bases. All displayed distributions can be modeled using a log-normal distribution. By varying the model parameters, we can simulate unobserved, counterfactual read-target combinations. We then used these pairs to assess the false-negative rate of target detection (see results). Actual and simulated sequence length distributions were near-identical, suggesting that the results derived from the simulation are realistic.

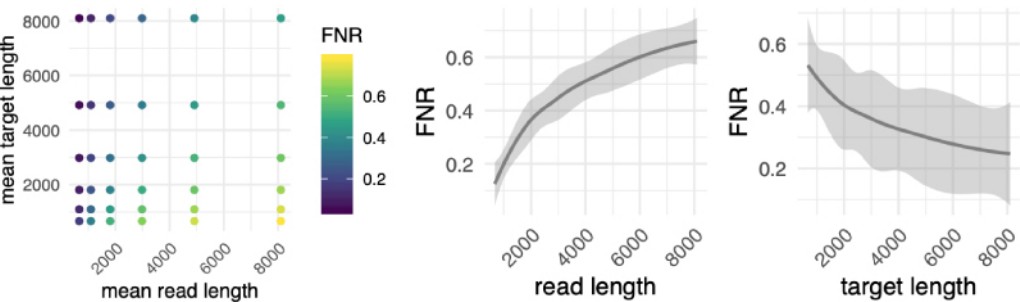

**Figure 3.** Simulation of counterfactual read-target pairs for various length distributions showing the large effect on the number of false-negative read rejections. The shorter the target relative to the median read length of the sequencing run, the larger the FNR (left panel, see results). For the use case discussed in this manuscript, namely the highly multiplexed detection of antimicrobial resistance genes, it was beneficial that the median read length matched the target size. A multivariate regression on the simulated pairs estimated this effect in further detail (middle and right panel).

remaining metagenomic contigs showed putative transfers. Again, metagenomics did not add important information beyond the culture isolates.

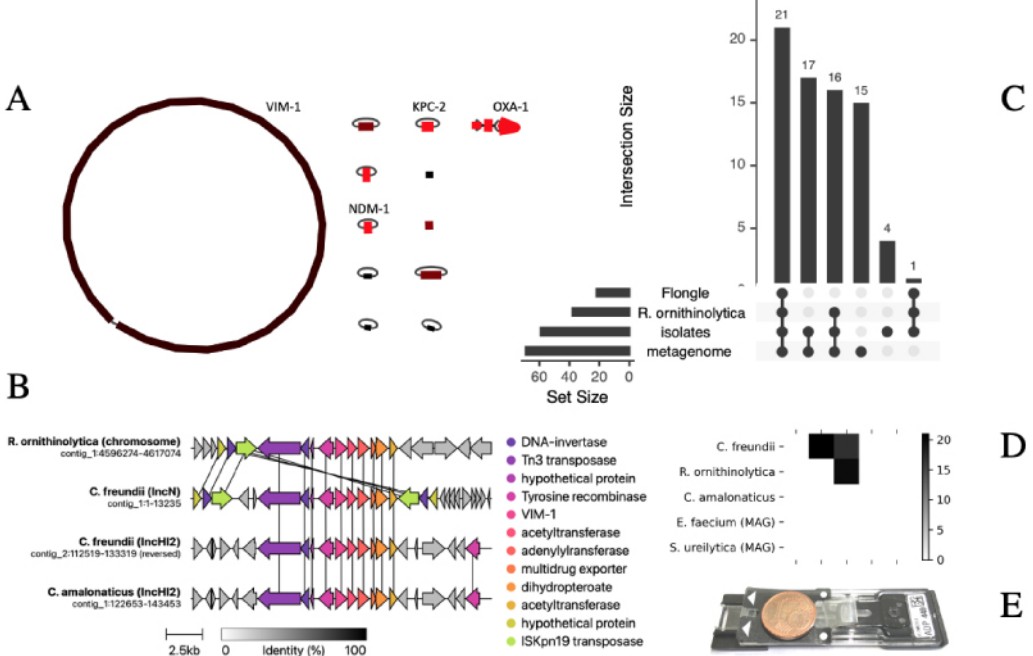

**Figure 4.** Real-time sequencing revealed extensive resistance load and horizontal gene transfer. (A) Genome reconstruction of a strain of *R. ornithinolytica* carrying nine plasmids and three carbapenemase genes (in addition to two linear plasmid fragments mapping to *R. ornithinolytica* plasmid `MT062911.1`, NCBI). Color-coded coverage from 90× (black, e.g., chromosome) to 250× (red, e.g., plasmid carrying *OXA-1*). (B) Gene transfer of *VIM-1* across three strains and four loci. The carbapenemase is flanked by multiple transposases (see annotation), which likely mediate its mobilization. Vertical lines indicate 100% sequence identity between corresponding genes. (C) Comparison of shared resistance genes between the enrichment sequencing run ("Flongle"), the *R. ornithinolytica* isolate, all four "isolates" combined and the "metagenome" assembly. Numbers correspond to absolute numbers of (shared, de-duplicated) genes. Of all resistance genes identified in the metagenome, 85.5% (59/69) were found in the isolates. Surprisingly, several resistance genes were not identified in the metagenome, among them several carbapenemase copies. In the *R. ornithinolytica* isolate genome, about two-thirds of the resistance genes were also found using on-device target enrichment. All plasmid-encoded genes among them were detected, including all carbapenemases. (D) Pairwise shared sequences between isolates and metagenome-assembled genomes (the grayscale represents the number of shared sequences). Putative transfers were defined as loci with a minimum length of one kilobase and 99.9% sequence identity between each pair of loci. Extensive sequence transfer was observed between the three isolate genomes (and their corresponding bins from the metagenomic assembly). (E) Miniature, low-cost flow cell used for on-device target enrichment ("Flongle", Oxford Nanopore Technologies), with a one-cent coin placed on top as scale.

## Adaptive sequencing effectively enriches for antimicrobial resistance genes

The sensitivity of metagenomic sequencing can be increased with depth, but the associated cost limits the applicability in the laboratory routine. Therefore, going in the opposite direction, the "Flongle" nanopore flow cell aims to reduce per-run costs through reduced sequencing yield. Because the yield is reduced, however, targeted sequencing of relevant genes or loci is desirable. Such target enrichment can be performed "on-device", i.e., during the sequencing run in real-time and without any changes in the sample preparation, using a method also known as nanopore "adaptive sequencing" [24–26]. Here, reads were rejected from the pore when the read fragment that already passed through it did not match any sequence in a target database. The nanopore was then free to sequence another molecule.

Adaptive sequencing can be used to enrich or deplete either entire organisms from a sample of DNA or to target specific genes [25, 26]. Here we aimed to enrich 1,147

representative antimicrobial resistance genes (ARGs, see methods). To our knowledge, this is the first time that adaptive sequencing has been used to target a microbial gene panel. We defined *enrichment* as the difference in total read count over a corresponding ARG between standard and adaptive sequencing. In the latter condition and unless stated otherwise, we excluded rejected reads, i.e., those for which the adaptive selection algorithm had not recognized a target in the first bases of the read (Figure 2).

Within the score of the presented use case, our enrichment definition makes sense because our aim was to count (and optimize for) the absolute number of detected resistance gene copies. Analogous to digital PCR, an increased number of detected copies translates into an increased test sensitivity. This definition should not be applied when the aim is, e.g., to balance the target coverage. For example, Payne *et al.* demonstrated how adaptive sampling could be used to balance the sequencing coverage for an unbalanced, mixed culture of microbial species [26]. In their enrichment protocol, reads from all genomes are initially accepted until a target coverage is reached, at which point reads from the associated genome are rejected. This protocol effectively shifts the number of bases sequenced towards less-covered genomes because rejected reads are much shorter, while the proportion of reads remains the same (a result we replicated, see below). Thus, one can still determine the relative abundance during such an experiment.

However, detecting and comparing the number of gene copies between conditions is a wholly different use case. Read count over target genes is a commonly used metric in RNA-Seq experiments, with two caveats: First, most RNA-Seq studies use fixed-sized short reads, i.e., the read length distribution is the same for all conditions. Second, the read count is normalized by target length and the overall number of mapped reads, which allows the comparison of targets *within* a condition ("relative expression"), although their size might differ. Here, we only used the raw read count to measure the enrichment without further adjustments because of the following reasons. First, a single library (and thus a single pool of DNA fragments) was used in both conditions (standard, adaptive), and the resulting read length distributions for "standard" and "unrejected adaptive" reads were near-identical (Figure 2). Second, we only compared each target gene *between* the two conditions, so target length normalization was not required.

To compare the target read abundance between adaptive and "standard" nanopore sequencing, we sequenced three isolates in technical duplicates on a single MinION flow cell, periodically alternating between adaptive and standard sequencing in 16 one-hour intervals (Figure 5A). Overall, adaptive sequencing could roughly double the abundance of many targets, while others were hardly detected (Figure 5B). We subsequently identified two factors that substantially affected the target abundance between the two conditions: nucleotide identity and read length.

First, high nucleotide identity between an isolate's gene and the corresponding member of the target gene panel resulted in a higher on-target read count (Figure 5B). Surprisingly, the most similar and, thus, the most enriched genes were located on plasmids. This likely reflects a bias in the database composition, where common resistance plasmids are well annotated while strain-specific, chromosomal gene isoforms are undersampled. As expected, target sequence similarity did not affect abundance in the standard condition, which did not use a target database. To quantify this effect, we performed Bayesian regression and modeled the effects of variables for which a contribution to abundance seemed plausible, namely sequence similarity, coverage, read length and whether the target

was located on a plasmid (including interaction effects, see methods). The largest effect was observed for similarity conditional on whether adaptive sequencing was turned on ($\beta$ = 11.98, 95% *CI* ± 0.15). However, it can be hard to interpret any single coefficient in an interaction model in isolation; it is more informative to plot samples from the posterior distribution for any variable of interest. All else being equal, adaptive enrichment only outperforms standard sequencing when an isolate's gene has a nucleotide identity of at least 95% to a record in the target database, and up to two-fold for near-identical targets (Figure 5C). Several targets are enriched four times over the standard baseline. Other studies reported a similar enrichment of two- to four-fold for bacterial genomes, albeit partly using different real-time matching algorithms [25, 27].

Second, we observed that reads from plasmids were shorter than chromosomal ones (Figure 5D). Furthermore, the length of chromosomal reads is shorter for adaptive sequencing than the standard because if a target is not identified on a given read, sequencing is terminated prematurely. Our statistical model takes this conditionality into account. The model indicates that adaptive sequencing outperforms the standard only for reads smaller than 3 kb, all else being equal (Figure 5E). This bias contributes to the higher target abundance for plasmid-encoded genes than chromosomal ones within the adaptive sequencing condition.

It seems counterintuitive that short reads are enriched more than longer ones. However, this is due to how adaptive sequencing rejects reads. The algorithm scans the first part of each read for target sequences, and if none is found after several hundred bases, the read is rejected (median 415 bases, Figure 2). To further investigate the relationship between read length, target length, and false-negative rate (FNR), we simulated combinations of, e.g., long reads with short targets and short reads with long targets (see methods). A fixed-length target occupies a smaller fraction in long reads than in shorter ones. We found that the smaller this fraction, the higher the FNR, i.e., the more reads which contain the target are falsely rejected (Figure 3). The intuition behind this result is that a target has many potential starting points on a read. The longer the read and the shorter the target, the more likely the target starts at a position after the read interval that the selection algorithm uses for its decision.

The relative abundance of reads *outside target regions* should not change during adaptive sequencing [26]. We checked this by comparing the coverage of "standard" and "rejected adaptive" reads across assembly contigs, which did not differ substantially (Figure 6). Note that we normalized the coverage to that of the chromosome for both read sets because the rejected reads from the adaptive condition were much shorter and more numerous (Figure 2, median read length 3,194 vs 415 bases).

## Adaptive sequencing detects all clinically relevant resistance genes on a miniature flow cell

We then tested the enrichment on a type of miniature, low-cost flow cell ("Flongle") and generated 5.4 Mb within four hours from the carbapenemase-rich *R. ornithinolytica* isolate (Figure 4E). 97.2% of the reads were rejected; of these, 0.2% (*n* = 43) were false negatives. Correspondingly, 2.8% of the reads were accepted, of which 20.4% (*n* = 104) were true positives, i.e., could be found in the target database. A positive database hit was defined as a read with at least 100 bp mapped to a target with a minimum of 50% matching positions. 57.9% (22/38) of the resistance genes found in the high-quality genome reconstruction were

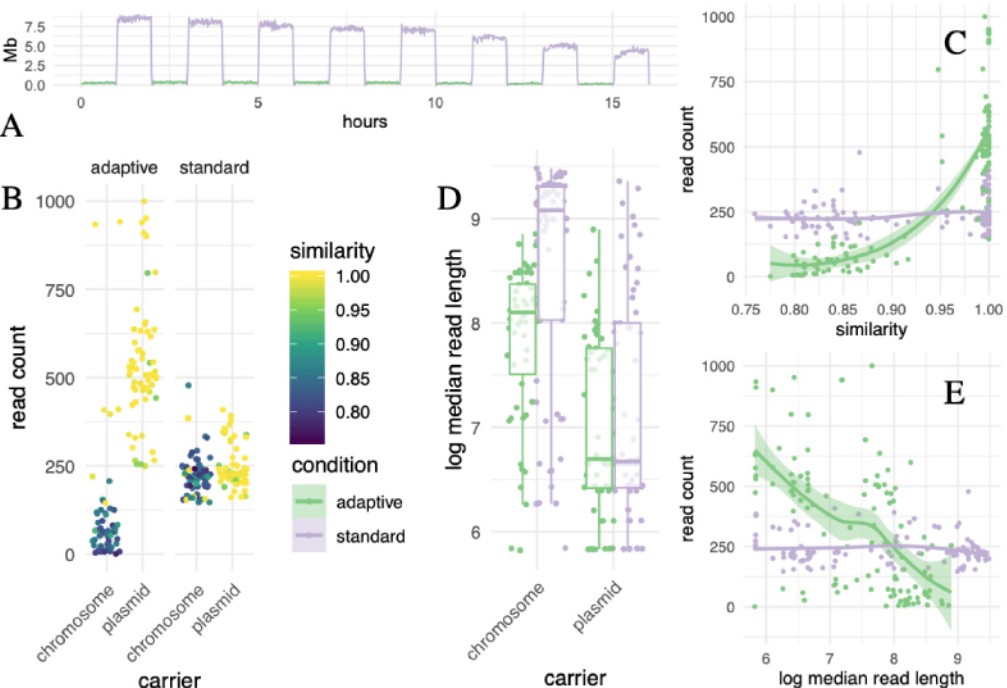

**Figure 5.** Effect of two variables during *adaptive* sequencing on enrichment efficiency compared to a *standard* nanopore sequencing run. (A) Setup of the sequential experiment to compare adaptive and standard sequencing, with the same samples on the same flow cell. Three *Citrobacter* and *Raoultella* isolates were sequenced with and without enrichment, alternating between these two conditions every hour on a single flow cell (green is adaptive sampling turned on, violet is standard sequencing). Note how the average sequencing yield per minute in megabases (y-axis, rolling mean) differed across time (x-axis): in the adaptive sampling condition, most reads were rejected after around 400 bases, which, as expected, resulted in a substantially reduced total yield in this condition. (B) Each point corresponds to an open reading frame that was annotated in the final isolate genome assembly as a resistance gene using a dereplicated ARG database ($n = 1,147$). "Read count" is the number of reads from each sequencing condition that mapped to these genes. As a read passed through the nanopore during enrichment, it was searched in real-time against a database of target genes. If no match was found, the read was ejected prematurely. Reads that were *similar* to a database entry (nucleotide identity) passed this filter, while reads with lower sequence identity were falsely rejected. Many highly similar targets resided on plasmids, likely a sampling bias in the resistance database. As expected, similarity had no effect on read count per open reading frame for standard sequencing because there was no database search involved. (C) Adaptive sequencing outperformed the standard once the nucleotide identity ("similarity") between the target and its match in the panel database surpassed 95%. For values close to identity, a two-fold enrichment can be expected. We even observed a four-fold enrichment of several targets over the standard baseline. When we fitted a Bayesian multivariate regression model, the increase of target abundance with similarity became clear (2.5 and 97.5% quantiles displayed). (D) Median read length in bases (y-axis) was plotted using a log scale for visual clarity and compared between chromosomal and plasmid DNA (x-axis). As orientation, the logarithm of one thousand (bases) is about 7, the logarithm of ten thousand is around 9. Reads derived from plasmids are shorter than chromosomal ones. In turn, chromosomal reads from adaptive are shorter than those from standard sequencing because they are more frequently ejected from the nanopore before the read has been sequenced fully for lack of any match. These factors need to be accounted for in a regression model to estimate the effect of read length on target abundance accurately. (E) Adaptive sequencing outperformed the standard for read lengths of 3 kb and less, all else being equal (for visual clarity, the read length is logarithmic as in (D), and 3 kb corresponds to a logarithm of about 8, where the green and violet lines intersect). Short reads of about 1 kb (log 6) can potentially double the target abundance. Therefore, library preparation protocols for adaptive sequencing could add a step to shear the extracted DNA to improve the enrichment.

found using adaptive sampling, too, including all three carbapenemases (Figure 4C). As expected from the adaptive-standard state switching experiment, the probability of detection was determined by genomic location: all the un-detected genes were located on the chromosome, and all the plasmid-encoded resistance genes were detected (odds ratio 26.7, $p < 0.001$). While plasmids are present in higher copy numbers relative to the chromosome (Figure 4A), our regression model did not assign a large effect to this variable.

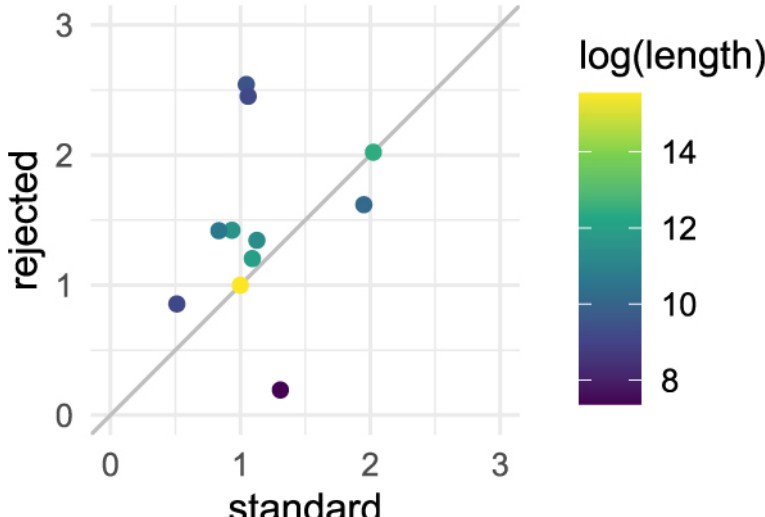

**Figure 6.** Relative coverage of reads generated using standard sequencing and the rejected read fraction from adaptive sequencing (isolate A2). Adaptive sequencing has been shown not to change the relative abundance of reads *outside target regions* [26]. Correspondingly, we expected the same coverage from both groups, which we found, validating the correctness of the adaptive sequencing run. Note that we normalized the coverage of all contigs within each condition to the coverage of the chromosome because the rejected reads were more numerous and shorter. Comparing unadjusted coverages would artificially inflate the coverages from the rejected reads. The normalized coverage was similar between standard and rejected reads, except for the smallest contigs. Here we saw a slight deviation that, due to the small contig size, did not affect the conclusions of this study.

Since many resistance determinants are located on plasmids, we argue that enrichment sequencing is a promising approach for antimicrobial gene detection in routine settings. Note, however, that the sequencing yield from Flongle flow cells was still more variable than for other flow cell types at the time of writing. While the Flongle yield presented here is typical in our experience, we occasionally see unusual deviations for no apparent reason. This variability might make adoption in routine workflows more challenging compared to other flow cell types.

## DISCUSSION

We detected a highly resistant consortium during hospital admission screening, including a strain that carried three carbapenemases. Nanopore sequencing comprehensively characterized three resistant culture isolates, documenting many resistance genes as well as extensive gene transfer between isolates. The metagenomic sequencing of the corresponding rectal swab added little information and did not detect several important resistance genes. It might be that a deeper sequencing would increase the sensitivity. Still, because the carbapenemase-carrying strains were low abundant, in practice, this procedure would not be cost-competitive in a routine setting. Cultural screening, as a first step, reliably identified the strains that carried clinically relevant resistance genes within 24 hours from sample streaking on screening agar plates to detectable growth. From the subsequent sequencing, including library preparation to isolate genome assembly, another 24 hours passed. This short turn-around time helped shape the public health response. For example, transposon-encoded *VIM* and *OXA* meant that associated wards could be monitored for the occurrence of these genes in other members of the *Enterobacteriaceae*. By comparison, generating 20 Gb of metagenomic data would also take two days, irrespectively

of the sequencing platform. In addition, more computation is required for the final assembly, binning, and validation of the mixed sample. We recently explored a combination of metagenomic sequencing with adaptive sampling, which could, in principle, further reduce the time to generate the results [28]. However, we observed a moderate enrichment by about a factor of two, which would roughly halve the cost of metagenomic sequencing. While this is still about an order of magnitude too expensive for most routine use cases, moderate decreases in sequencing costs combined with better enrichment through better sample preparation and adaptive sampling efficiency could approach this threshold soon.

We then evaluated a new approach for on-device, real-time target enrichment called "adaptive sequencing". It encompassed 1,147 representative antimicrobial resistance genes in an ultra-high multiplex assay. In the enrichment sequencing data, all and 57.9% of the known resistance genes from matched isolate assemblies were identified on a MinION and Flongle flow cell, respectively. However, while some genes could be enriched up to four times over the baseline, others were hardly captured. To explain this disparity, we found that two variables influence the enrichment substantially.

First, the higher the nucleotide similarity of a read to its corresponding entry in the target database, the more reads were selected. Adaptive outperformed standard sequencing above 95% identity. The optimization of the target database to reflect the expected targets as closely as possible is thus crucial. Future studies will have to determine the influence of database size and redundancy on target abundance. However, the sequence identity of a given read to a database target has two sources. One is the already discussed biological variability, for which the experimenter can adjust the database composition. The second source is sequencing error. Any improvement to the nanopore technology to reduce per-read sequencing error, from pore shape to basecalling algorithms, would likely improve the adaptive sequencing results further.

Second, we showed that fragments shorter than 3 kb are beneficial to the target abundance. Counterintuitively for nanopore sequencing, deliberate shearing of DNA fragments during library preparation should help to enrich targets. From an economic perspective, the fold-enrichment is inversely proportional to the sequencing cost. Hence, an enrichment by a factor of two would translate into 50% reduced sequencing costs (excluding library preparation). Alternatively, more enrichment leads to a shorter time-to-answer or increases the sensitivity of the assay, whatever the researcher prioritizes in their use case.

Regarding the adaptive sequencing experiment setup, it may have been better to run adaptive sampling on half the channels continuously rather than stopping and starting the sequencing run every hour. However, while more convenient to the user because it does not require manual mode switching, this might have introduced bias if the channels in one split were less active than in the other. Either way, the conclusions drawn are unlikely to change substantially between the sequential and parallel setup of the experiment.

The degree to which DNA should be sheared for an enrichment experiment depends on the underlying choice of target length. For the enrichment of antimicrobial resistance genes, as demonstrated in this study and with a mean length of about one kilobase, we suggest matching this with an equal median read length. Note that this suggestion is based on modeling results derived from limited experimental data (one isolate, eight on-off cycles of adaptive sequencing). Future work will have to demonstrate the effectiveness of these optimizations under a broader range of isolates and sample conditions.

On-device target enrichment could be demonstrated for the most resistant culture isolate on a Flongle flow cell. Also, we were able to identify all the plasmid-encoded resistance genes and more than half of all the resistance genes known to be present. This proof-of-principle motivates further exploration of the clinical utility of this sequencing protocol and hardware.

## CONCLUSIONS

We show that on-device target enrichment on low-cost flow cells could be a valuable complement to routine microbiology. It takes us closer to an effective point-of-care resistance screening, especially given the continuing rapid improvements in the underlying technology [29]. However, given the variable sequencing yield of this new flow cell type, further controlled experiments that compare multiple runs with and without enrichment are warranted, as are studies that optimize sample preparation and target database composition.

## LIST OF ABBREVIATIONS

ARG: antimicrobial resistance gene; FNR: false-negative rate; MAG: metagenome-assembled genome.

## ETHICS APPROVAL AND CONSENT TO PARTICIPATE

Not applicable; only microbial samples were used, which are not subject to ethical approval. Human DNA sequences were removed from the metagenomic stool dataset before analysis by filtering them against the recently published complete human reference genome CHM13 [19].

## CONSENT FOR PUBLICATION

Not applicable. The manuscript includes no specific details, images or videos relating to an individual person.

## DATA AVAILABILITY

All basecalled nanopore sequencing data has been deposited with the SRA, NCBI. Metagenomic reads are available under project ID PRJNA788147. Reads from isolate genomes, the Flongle flow cell and the experiment alternating between adaptive and standard state have been deposited under project ID PRJNA788148 under their respective sample ID (*Raoultella ornithinolytica*: SAMN23928631, *Citrobacter freundii*: SAMN23928632, *Citrobacter amalonaticus*: SAMN23928633).

Beyond standard analyses described in the methods, code and referenced assemblies (isolates, metagenome) for the following analyses are available from a dedicated code repository at https://www.github.com/phiweger/adaptive: Putative horizontal gene transfer, creation of the target database for adaptive sequencing, analysis of the experiment that switched adaptive sequencing on and off and how to specify a Bayesian regression model of the target count. A snapshot of the code is also available in the GigaDB repository [30].

## DECLARATIONS

### Competing Interests

AV has received travel expenses to speak at Oxford Nanopore meetings. AV, CB and MH are co-founders of nanozoo GmbH and hold shares in the company.

## Funding

None received.

## Authors' Contributions

AV designed the study. AV and ND collected and characterized all samples. MM and CB extracted DNA, prepared Nanopore libraries, and performed all sequencing experiments. AV, CB, and MH analyzed the data. All authors revised the manuscript critically and approved the article's final version for publication. MWP and CB supervised the study.

## Acknowledgements

We thank all technical assistants who supported laboratory tasks.

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
