## [Reviewer Report]

Reviewer name and names of any other individual's who aided in reviewerJulian SommerDo you understand and agree to our policy of having open and named reviews, and having your review included with the published manuscript. (If no, please inform the editor that you cannot review this manuscript.)YesIs the language of sufficient quality?YesPlease add additional comments on language quality to clarify if neededIs there a clear statement of need explaining what problems the software is designed to solve and who the target audience is? NoAdditional CommentsNot applicable to this study, since no novel software is described.Is the source code available, and has an appropriate Open Source Initiative license <a href="https://opensource.org/licenses" target="_blank">(https://opensource.org/licenses)</a> been assigned to the code?YesAdditional CommentsAs Open Source Software are there guidelines on how to contribute, report issues or seek support on the code?YesAdditional CommentsIs the code executable?YesAdditional CommentsThe code used for analysis of the data has been published on the corresponding github page. Although, a link on this page for downloading data from a public database does not work at the time of testing. (Resource deleted). Also, most parts of the code are executable, the generated data and figures resulting from the code does not reporduce the figures from the publication.Is installation/deployment sufficiently outlined in the paper and documentation, and does it proceed as outlined?YesAdditional CommentsThe code placed in the github repository can be executed mostly, but require basic knowledge of coding in the used programming languages. However, for the data presented in this work, I do not see the need for more detailed instructions.Is the documentation provided clear and user friendly?YesAdditional CommentsOnly partlyIs there enough clear information in the documentation to install, run and test this tool, including information on where to seek help if required?YesAdditional CommentsOnly partlyIs there a clearly-stated list of dependencies, and is the core functionality of the software documented to a satisfactory level?NoAdditional CommentsOnly partly. However, I do not see the need for further instructions.Have any claims of performance been sufficiently tested and compared to other commonly-used packages? NoAdditional CommentsIs test data available, either included with the submission or openly available via cited third party sources (e.g. accession numbers, data DOIs)?YesAdditional CommentsThe data is available from the stated accession numbers, but an additional data link on the github page does not work and might be necessary to test the complete code.Are there (ideally real world) examples demonstrating use of the software? YesAdditional CommentsIs automated testing used or are there manual steps described so that the functionality of the software can be verified?NoAdditional CommentsAny Additional Overall Comments to the AuthorThe study compared three methods of oxford nanopore-based longread sequencing for detection of antibiotic resistant bacterial pathogenes. Therefore, the authors used cultivation based detection of carbapenem-resistant bacteria from a rectal swap and subsequent singe isolate sequencing. This technique was compared to an adaptive sequencing approach using a database of antibiotic resistance genes for adaptive sequence enrichment during the sequencing run facilitation oxford nanopore sequencing. The underlying technology is a unique approach, made possible by the oxford nanopore real-time sequencing technology and is of great interest for future applications in clinical microbiology diagnostics. Therefore, this study is of great importance for this field in general.  As additional method, the authors performed metagenome sequencing of the rectal swap without culture, which is a completely different technique with unique advantages and drawbacks, compared to culture-based sequencing methods.  This study is important for the development of real time sequencing and adaptive sequencing for the detection of antibiotic resistance genes and in future potentially other genes. It focusses on the adaptive sequencing approach, analysing in detail the factors influencing the performance of this new approach. The number of experiments is limited, as stated by the authors, but the data is nevertheless valuable for future projects. For further improvement, I have some suggestions for the manuscript. 1. The comparison of the three methods is quite complex and one of the main goals of this paper, illustrating, that low-cost sequencing devices (Flongle) can be used for detection of antibiotic resistance genes applying adaptive sequencing. Therefore, the description of this comparison and figure 1C is essential for understanding the data of this comparison of methods. However, figure 1C is hard to read and the represented data is not easily accessible. To clarify, I suggest including additional information. Does the “Set size” and “Intersection Size” describe absolute number of detected antibiotic resistance genes? This information could be included. To achieve additional connection from the legend of figure 1C, the absolute numbers of detected genes could be included to the text, supplementing the already stated relative detection numbers (lines 51-54, 137-142). Since this figure part is essential for the understanding, a larger version of this representation would be nice. 2. Figure 2 is essential for interpretation of the presented data on variables influencing the adaptive sequencing performance.  a. Figure 2A is not easily accessible, in fact I am not sure, what information about the data is represented in this part of the figure (data throughput?). The figure legend does not explain, what is shown. I suggest clarification or, if applicable, deletion of this subfigure, for increased readability of figure 1B-D. b. Figure 2D: The meaning of the “log median read length” is not explained in the text or the figure legend and should be clarified.  c. Figure 2E: Same as for Figure 2D. In line 119, the absolute read length (3 kb) is stated, but this number is not visualised in this figure. I suggest adding additional information to the text, to make the representation of the data in the figure easily discoverable. 3. Discussion: In my opinion, the discussion part has some potential for improvement. a. Line 158 – 162: The authors argue that selective cultivation and subsequent adaptive sequencing for antibiotic resistance genes leads to rapid results, important for public health responses. Metagenomic sequencing on the other hand needs at least the equal time and is not cost effective. However, might the combination of metagenomics sequencing without culture and adaptive sequencing decrease the turnaround time even more without significantly higher costs? Although, experiments on this are not in the scope of this study, the authors could discuss this for future applications. b. Line: 165: “[…] reads were detected for all resistance genes known to be present […] This result does not match the results stated in line 141 “57.9 % of the resistance genes found” and line 184 “nearly two-thirds of all resistance genes”. This should be clarified or the corresponding data should be referenced in the discussion for readability. c. Line 169: Since the identity of sequencing results and hit to the database is important for detection and overall performance of the adaptive sequencing approach, I suggest discussing, if future improvement of sequencing accuracy (basecalling algorithm, pore design) might influence the performance of this approach, as only shortly mentioned in line 190. d. Line: 190 “variable sequencing yield of this new flow cell type”: This aspect is solely introduced in the conclusion and should be mentioned and discussed beforehand.     Minor comments: 1. Figure 1 description: “[…] carrying nine plasmids and four carbapenemases genes […]”. In line 12, the Raoultella isolate is described carrying three carbapenemases. The OXA-1 beta-lactamase pictured in figure 1A is not a carbapenemase. The correct number should be three carbapenemases. 2. Line 67: Flongle flowcells were introduced in 2019. I suggest to delete “recently introduced”. 3. Line 210: The link is not correct. 4. Line 244: “Community standards”: It would be nice to add an additional reference. 5. Line 255. Reference is missing. 6. Line 283: This step RecommendationMinor Revisions

---

## [Reviewer Report]

Upload additional filesTRR-202211-03/form/comments.docxReviewer name and names of any other individual's who aided in reviewerNed PeelDo you understand and agree to our policy of having open and named reviews, and having your review included with the published manuscript. (If no, please inform the editor that you cannot review this manuscript.)YesIs the language of sufficient quality?YesPlease add additional comments on language quality to clarify if neededIs there a clear statement of need explaining what problems the software is designed to solve and who the target audience is? YesAdditional CommentsIs the source code available, and has an appropriate Open Source Initiative license <a href="https://opensource.org/licenses" target="_blank">(https://opensource.org/licenses)</a> been assigned to the code?YesAdditional CommentsScripts have been made publicly available on GitHub (https://www.github.com/phiweger/adaptive) under an OSI-approved BSD-3-Clause license.
As Open Source Software are there guidelines on how to contribute, report issues or seek support on the code?NoAdditional CommentsIs the code executable?Unable to testAdditional CommentsIs installation/deployment sufficiently outlined in the paper and documentation, and does it proceed as outlined?YesAdditional CommentsIs the documentation provided clear and user friendly?YesAdditional CommentsIs there enough clear information in the documentation to install, run and test this tool, including information on where to seek help if required?NoAdditional CommentsIs there a clearly-stated list of dependencies, and is the core functionality of the software documented to a satisfactory level?NoAdditional CommentsHave any claims of performance been sufficiently tested and compared to other commonly-used packages? Not applicableAdditional CommentsIs test data available, either included with the submission or openly available via cited third party sources (e.g. accession numbers, data DOIs)?YesAdditional CommentsAre there (ideally real world) examples demonstrating use of the software? YesAdditional CommentsIs automated testing used or are there manual steps described so that the functionality of the software can be verified?YesAdditional CommentsAny Additional Overall Comments to the AuthorRecommendationMinor Revisions